# Comparison of NADAL COVID IgG/IgM rapid test and DiaSorin Liaison SARS-CoV-2 S1/S2 IgG assay across different blood sources and substrates

Harika Dasari,[1,2] Claude Bourassa,[3] Christian Renaud,[3] Cécile Tremblay,[4] Francine M. Ducharme[1,5]

**ABSTRACT**    The NADAL COVID-19 IgG/IgM (NADAL) assay, approved for rapid testing on serum or plasma, had not been approved nor tested as a point-of-care test on capillary blood. The study aim was to evaluate the performance of NADAL compared to that of the DiaSorin Liaison SARS-CoV-2 S1/S2 IgG (DiaSorin) assay (gold standard). We conducted two cross-sectional studies in participants aged ≥2 years, in whom different blood substrates (whole blood vs serum) and sources (venous vs capillary) were sampled. The co-primary endpoints were agreement in IgG detection between NADAL on venous serum and whole capillary blood vs DiaSorin on venous serum. Among 128 participants, 23.2% had a positive DiaSorin IgG assay. The NADAL IgG on venous serum exhibited near-perfect agreement (κ = 0.91) and high accuracy (0.94, 95% CI: 0.87–1.00), with a sensitivity of 0.88 and perfect specificity compared to DiaSorin; NADAL IgG on whole capillary blood exhibited moderate agreement (κ = 0.77) and accuracy (0.84, 95% CI: 0.73–0.95), with a sensitivity of 0.68 and perfect specificity. As for secondary outcomes, when the two assays were compared within the same source (venous or capillary) but with different substrates (NADAL on whole blood vs DiaSorin on serum), strong agreement and high accuracy were observed. Within-test DiaSorin IgG assay on capillary vs venous serum showed near-perfect agreement and high accuracy. Within-test NADAL IgG assays on whole capillary blood vs capillary or venous serum demonstrated high accuracy. In conclusion, NADAL on venous serum exhibited near-perfect agreement and high accuracy with DiaSorin on venous serum but lower between-test performance on different blood substrates.

**IMPORTANCE** Accurate serological assays are essential for assessing population immunity levels and identifying vulnerable subgroups with lower immunity on which to focus vaccination efforts. Although rapid tests may offer the possibility of easy point-of-care testing on whole capillary blood for these purposes, they may not be as robust as other assays when used on different blood substrates or sources than those in which they were approved. The observed variability in test performance across different substrates and blood sources highlights the importance of pretesting assays under the specific conditions in which they would be used to ensure optimal interpretation of the immune status in the community.

**KEYWORDS**    NADAL COVID-19 Test, SARS-CoV-2 IgG/IgM serology, sensitivity and specificity, whole capillary blood, point-of-care testing, diagnostic accuracy

The coronavirus disease 2019 (COVID-19) pandemic related to the severe acute respiratory syndrome coronavirus 2 (SARS-CoV-2) caused unprecedented challenges, and the ongoing emergence of new variants remains a significant concern worldwide (1). Detecting infected individuals quickly and ascertaining the population immune

**Peer Reviewer** Sara Assadiasl, Tehran University of Medical Sciences, Tehran, Iran

Address correspondence to Harika Dasari, Harika.dasari@umontreal.ca.

F.M.D. has received unrestricted research funds from AstraZeneca, Covis Pharma, GlaxoSmithKline, Merck Canada, Novartis, Teva, Trudell Medical, GlaxoSmithKline, and MEDteq in partnership with Thorasys Inc.; an honorarium for consultancy work from AstraZeneca, Covis Pharma, Sanofi, Teva, and Thorasys Inc.; and an honorarium as an invited speaker from Covis Pharma, Jean-Coutu Pharmacy, and Brunet Pharmacy. All other authors have no conflict of interest.

See the funding table on p. 11.

status accurately have been crucial in controlling the spread of infection (2). During the COVID-19 pandemic, the detection methods for SARS-CoV-2 infection have evolved. Due to its high sensitivity and specificity, the quantitative reverse transcription-polymerase chain reaction (RT-qPCR) tests are the gold standard in confirming SARS-CoV-2 infections (3), although less sensitive antigenic tests are quicker, cheaper, and, importantly, enable self-screening (4). Natural SARS-CoV-2 infection triggers antibodies to multiple viral components, including both the nucleocapsid (N) and spike (S) proteins, whereas COVID-19 vaccines are designed to elicit an immune response primarily against the S protein of SARS-CoV-2, such that anti-S serology testing captures both natural and vaccine-induced immunity (5). Consequently, serology tests that detect the presence of antibodies against SARS-CoV-2 have been crucial for disease surveillance at population levels and for examining the response to vaccination (6). Unlike antigen tests, which detect active infection, serology tests assess prior immune response. While their role has diminished after the pandemic, they will remain valuable, in future outbreaks, for monitoring seroprevalence in the general (overwhelmingly immunocompetent) population as well as the immune status in specific at-risk populations including immunosuppressed individuals. Yet, any rapid screening serology test must first be compared to a gold standard to ascertain performance properties before uptake.

Viral neutralization tests are considered the gold standard for antibody detection due to their accuracy, but they are not widely available (7, 8). Laboratory-based serology tests, including enzyme-linked immunosorbent assays (ELISA) and chemiluminescent immunoassays (CLIA), detect binding IgG and/or IgM antibodies to SARS-CoV-2 antigens. These assays have been developed to detect antibodies against the SARS-CoV-2 S and/or N proteins in blood samples (9, 10). Lateral flow immunoassays (LFIA) are point-of-care rapid antibody tests utilizing the lateral flow technology to detect IgG and IgM antibodies, providing quick and convenient results despite limitations in the quantification and potential for false positives (11, 12). Both the CLIA DiaSorin Liaison SARS-CoV-2 S1/S2 IgG, the first serology test for SARS-CoV-2 approved by Health Canada, and the LFIA NADAL COVID-19 IgG/IgM specifically target antibodies to the S protein (13, 14). The NADAL test qualitatively detects both IgG and IgM antibodies, and the DiaSorin Liaison provides quantitative IgG antibody titers (15). Comparing these tests using the same or different sample source (capillary or venous) and substrate (serum or whole blood) could ascertain their degree of concordance under different conditions.

Our co-primary objectives were to evaluate the test performance of the qualitative NADAL COVID-19 IgG/IgM LFIA, that is, its ability to detect IgG antibodies (positive, negative, or invalid) using both venous serum and whole capillary blood samples against the DiaSorin Liaison SARS-CoV-2 S1/S2 IgG quantitative assay on venous serum samples, serving as the reference. Secondary objectives examined the impact of various blood sources (venous/capillary) and their substrates (serum/whole blood) on the test performance of Nadal IgG alone or in combination with IgM. This study was conducted early in the pandemic (2020–2021), during a period of limited access to antigen testing, before widespread vaccination and during severe restrictions regarding in-person contact for venipuncture for research purposes. The study was designed as a proof-of-concept methodological evaluation of a point-of-care self-screening serology test on capillary blood to identify post-COVID infection in immunocompetent individuals.

## MATERIALS AND METHODS

We conducted two concurrent cross-sectional studies between 23 September 2020 and 6 July 2021 to assess the diagnostic performance of NADAL COVID-19 IgG/IgM compared to DiaSorin Liaison SARS-CoV-2 S1/S2 (reference) in individuals at risk of SARS-CoV-2 exposure. These studies were nested in an observational study testing a self-capillary blood collection device in children and adults (16), as well as in an adult randomized multicenter clinical trial entitled "Prevention of COVID-19 With Oral

Vitamin D Supplemental Therapy in Essential HealthCare Teams (PROTECT)" (17, 18). Both studies received approval from the Centre Hospitalier Universitaire Sainte-Justine (CHUSJ) Human Research Ethics Board (#2021–3067 and #MP-21–2021-3044, respectively). Health Canada provided a non-objection letter for using the NADAL COVID-19 IgG/IgM rapid serology test in the former study and an Investigational Testing Authorization (ITA) approval (application #322424) for the PROTECT trial (18). Participants provided informed consent, with assent obtained for children before their participation.

## Subjects

The observational study enrolled participants in two phases, first in individuals aged ≥12 years and older and then in children aged 1–17 years with their parents (16, 19). Those with (suspected or confirmed) active COVID-19 infection, recent travel, or otherwise high-risk contact with infected individuals were excluded in order to prevent contamination of study personnel as the study took place before widespread testing and vaccination efforts. In the PROTECT trial, adult healthcare workers aged <70 years residing in the greater Montreal area (Quebec, Canada) were targeted after the exclusion of those with vitamin D intake >400 IU/day, prior documented COVID-19 infection/vaccination, calcium/vitamin D disorder, active cancer, immunosuppressive medication, drugs altering vitamin D metabolism, prolonged absence from work/follow-up, or enrolment in a concurrent trial, as detailed elsewhere (18).

## Measurements

The SARS-CoV-2 S1/S2 IgG measured on venous serum on the DiaSorin liaison XL CLIA analyzer platform, hereafter referred to as "DiaSorin," served as the reference. The IgG antibody levels were quantified in arbitrary units (AU/mL), with an upper range limit of 400 AU/mL and an inter-assay coefficient of variation (CV) less than 5%; values ≥ 15.0 AU/mL were deemed indicative of a positive test (20). The NADAL COVID-19 IgG/IgM serology test, hereafter referred to as "NADAL," received approval from Health Canada for use on serum or plasma; with results available in 10 minutes, it could have the potential to be used as a point-of-care test on whole capillary blood samples. The kit, comprising the NADAL COVID-19 IgG/IgM cassette and a buffer solution, provides rapid qualitative results (i.e., antibody absence or presence) with excellent reported repeatability and reproducibility (>99% accuracy on 10 positive and 10 negative specimens) (14). The test's interpretation requires a positive enzymatic reaction (red line) in the control line "C"; the absence of such a line would signal an invalid test. The test is then interpreted as positive if a red line is also visible either in IgG and/or IgM zones or negative if otherwise (Fig. S1).

## Study outcomes

The two co-primary outcomes were the agreement in the detection of anti-SARS-CoV-2 IgG between the NADAL assay on two blood sources, namely, venous serum and whole capillary, and the DiaSorin on venous serum (reference), focusing on IgG to allow fair comparison with the DiaSorin. Other test performance metrics included test accuracy, sensitivity, specificity, positive predictive value (PPV), and negative predictive value (NPV). As a sensitivity analysis, co-primary analyses were repeated considering the combined results of NADAL IgG/IgM on the same blood sources, using the same metrics as stated above. Secondary outcomes included the agreement and test performance of the following: (i) DiaSorin in detecting SARS-CoV-2 IgG in capillary serum compared to DiaSorin on venous serum (reference); (ii) NADAL in detecting SARS-CoV-2 IgG in whole venous blood, compared to DiaSorin on venous serum (reference); (iii) NADAL in detecting SARS-CoV-2 IgG in whole capillary blood, compared to DiaSorin on capillary serum (reference); (iv) NADAL in detecting both IgG and IgM antibodies in whole capillary blood vs venous serum (reference), (v) whole venous blood vs venous serum (reference), and (vi) whole capillary vs capillary serum (reference).

## Procedures

Venous blood samples of 5 mL or more were collected by trained nurses through venipuncture, using gold cap vacutainers containing a silica clot activator and polymer gel for serum separation (Becton Dickinson Vacutainer Venous collection Tube; ref 3687986). A minimum of 400 µL capillary blood was collected primarily using the TASSO-SST self-sampling device (Tasso Inc, Seattle, USA) containing a thixotropic serum separator gel from either the upper arm (or lower back in young children) and occasionally via a contact-activated lancet (Becton Dickinson Microtainer). Whole capillary or venous blood specimens were either immediately tested with the NADAL or centrifuged at $2,000 \times g$ for 10 minutes to extract serum before freezing at −20C; processing occurred within 4 hours of collection, if collected onsite. Individuals who remotely self-collected capillary blood using the Tasso-SST device also self-tested remotely on whole capillary blood using the NADAL, under video guidance by a research assistant and using a Health Canada-approved study-specific step-by-step graphic brochure of the NADAL procedure. (Fig. S2) Batch analysis was conducted in the clinical laboratory for the DiaSorin and NADAL using freeze-thawed serum samples, extracted within 4 hours of collection following manufacturer guidelines (14, 21).

## Statistical analysis

Categorical variables were summarized as frequency and percentage; continuous variables were assessed for normality using the Shapiro–Wilk test. Normally distributed parameters are presented as mean ± standard deviation (SD), and non-normally distributed parameters are presented as median (25%, 75%). The within-patient agreement was assessed using Cohen's kappa coefficient (κ) ranging from −1 to 1, where 1 indicates perfect agreement, 0 indicates agreement equivalent to chance, and negative values indicate agreement less than chance (22); in the range observed in this study, we interpreted kappa as near-perfect (0.91–0.99), strong (0.80–0.90), and moderate (0.60–0.79) (23). To evaluate the test performance, we reported overall accuracy as the main performance indicator, calculated from the scalar approach based on the area under the receiver operating characteristic curve (AUROC) and presented sensitivity, specificity, and positive and negative predictive values (PPV and NPV) (24, 25). AUROC interpretation was categorized as follows: low (0.51–0.70), moderate (0.71–0.90), and high (>0.90) (26–28). We reported 95% confidence intervals (CIs) for key metrics. Differences in sensitivities between tests were statistically analyzed using a Z-test for independent proportions to determine whether test performance or agreement variations were significant, enhancing our understanding of each test's clinical utility. To further explore the between-test concordance between NADAL across the whole spectrum of antibody levels on the DiaSorin, we displayed log-transformed values and confidence intervals by NADAL binary results and conducted *post hoc* logistic regression models of continuous DiaSorin antibody levels by NADAL binary results. Statistical analyses were conducted using STATA version 15 (Stata Corp., College Station, Texas) and R version 4.2.1 (R Foundation for Statistical Computing, Vienna, Austria, 2022). All statistical tests were two-tailed, with significance set as $P < 0.05$.

## RESULTS

Between 23 September 2020 and 6 July 2021, we enrolled 128 individuals (36.7% male) with a median age of 32 years (range: 2 to 66 years) who participated in at least one comparison (Table 1). Of these, 112 individuals aged 12 years or more contributed data to at least one of the co-primary outcomes, including 90 and 86 participants from the observational study and the PROTECT trial, respectively (Fig. 1).

Among the 90 and 86 participants contributing paired data to co-primary outcomes, the prevalence, based on the detection of anti-SARS-CoV-2 IgG in venous serum using DiaSorin, was 26.7% and 22.1% for samples comparing it to the NADAL IgG on venous serum and whole capillary blood, respectively. Near-perfect agreement and high

**TABLE 1** Participant characteristics

| Characteristics | All participants ($N = 128$) |
|---|---|
| Age (years) | |
| Median (25%, 75%) | 32 (25.5, 41) years |
| Gender | |
| Male, n (%) | 47 (36.7%) |
| COVID status[a] | |
| Positive DiaSorin IgG, n (%) | (n = 112) 26 (23.2%) |
| Self-reported infection, n (%) | (n = 121) 13 (10.7%) |
| Vaccination Status[b] | (n = 31) |
| One dose, n (%) | 4 (12.9%) |
| Two doses, n (%) | 1 (3.2%) |

[a]Determined by the patient's self-reported history of COVID-19 infection confirmed by a positive SARS-CoV2 qPCR test.
[b]Number of COVID-19 vaccine doses received prior to the study visit. This information was gathered only for a subset of 31 participants.

accuracy (with high sensitivity and perfect specificity) were noted when comparing the DiaSorin and NADAL IgG tests, both using venous serum (Table 2). The NADAL IgG test on whole capillary blood showed moderate, yet lower kappa agreement and accuracy compared to the DiaSorin test on venous serum (reference); the sensitivity of NADAL on whole capillary blood (0.68) was lower than that on venous serum (0.88); however,

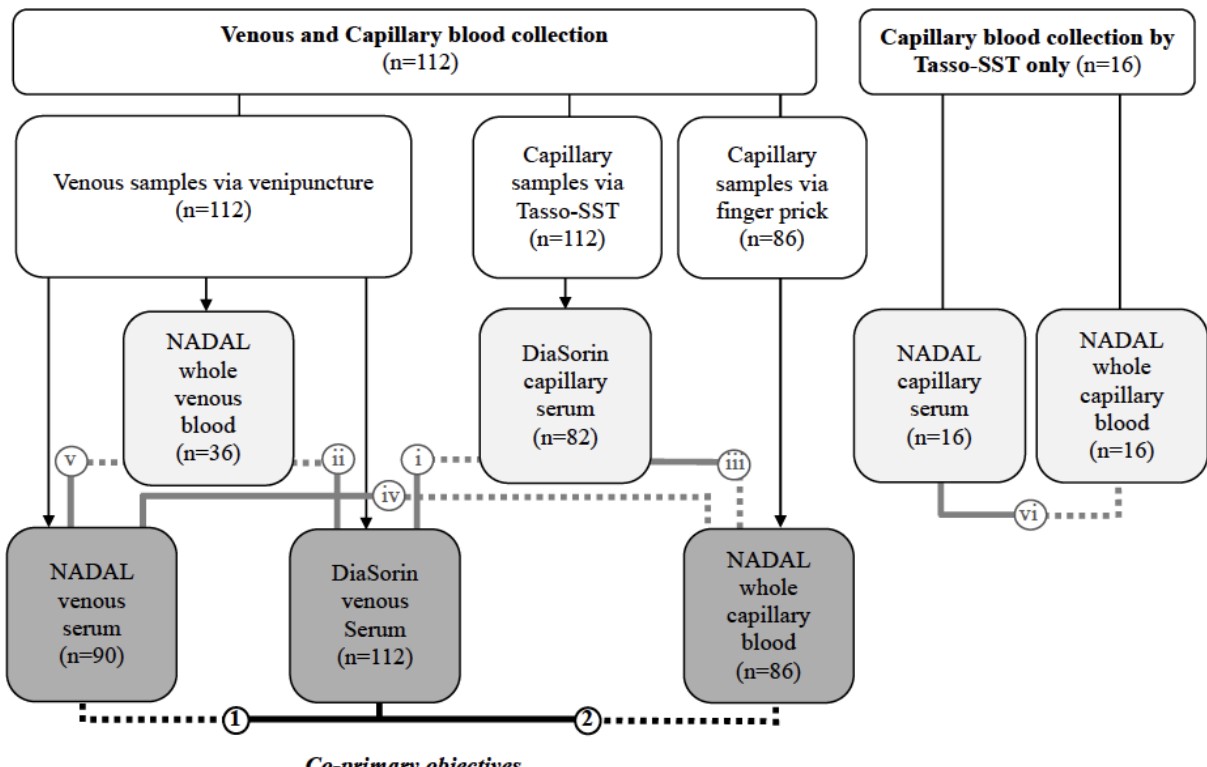

**FIG 1** Distribution of participants by comparison. As shown in the left panel, 112 individuals provided both venous (left middle box) and capillary (right middle box) samples. The capillary samples were obtained using the Tasso-SST self-sampling device (Tasso, Inc., Seattle, USA) from all 112 participants, of which 86 also provided additional capillary samples via finger pricking. The co-primary objectives compared IgG (and IgG/IgM) from the NADAL venous serum and whole capillary samples to DiaSorin venous serum (lower middle box). Secondary objectives included comparisons of (I) DiaSorin on venous serum vs DiaSorin on serum capillary blood; (ii) NADAL on whole venous blood vs DiaSorin on venous serum; (iii) NADAL on whole capillary blood vs DiaSorin on serum capillary blood; (iv) NADAL on venous serum vs NADAL on whole capillary blood; (V) NADAL on venous serum vs NADAL on whole venous blood; and (vi) NADAL on capillary serum vs NADAL on whole capillary blood. Dashed lines in the schematic represent the test conditions, while solid lines indicate the reference conditions.

**TABLE 2** Test performance and within-patient agreement of different serology tests across various blood sources and their substrates

| Reference[a] | Test[b] | Paired samples (N) | Reference[a] prevalence N (%) | Test[b] prevalence N (%) | Agreement κ (SE)[c] | P-value[d] | Accuracy[e] (95% CI) | Sensitivity (95% CI) | Specificity (95% CI) | PPV[f] (95% CI) | NPV[g] (95% CI) |
|---|---|---|---|---|---|---|---|---|---|---|---|
| DiaSorin IgG | NADAL IgG | | | | | | | | | | |
| Venous serum | Venous serum | 90 | 24 (26.7%) | 21 (23.3%) | 0.91 (0.11) | <0.001 | 0.94 (0.87–1.00) | 0.88 (0.68–0.97) | 1.00 (0.95–1.00) | 1.00 (0.84–1.00) | 0.96 (0.88–0.99) |
| Venous Serum | Whole capillary blood | 86 | 19 (22.1%) | 13 (15.1%) | 0.77 (0.11) | <0.001 | 0.84 (0.73–0.95) | 0.68 (0.43–0.87) | 1.00 (0.95–1.00) | 1.00 (0.75–1.00) | 0.92 (0.83–0.97) |
| DiaSorin IgG | NADAL IgG +IgM | | | | | | | | | | |
| Venous serum | Venous serum | 90 | 24 (26.7%) | 21 (23.3%) | 0.91 (0.11) | <0.001 | 0.94 (0.87–1.00) | 0.88 (0.68–0.97) | 1.00 (0.95–1.00) | 1.00 (0.84–1.00) | 0.96 (0.88–0.99) |
| Venous Serum | Whole capillary blood | 86 | 19 (22.1%) | 14 (16.3%) | 0.74 (0.11) | <0.001 | 0.83 (0.73–0.94) | 0.68 (0.43–0.87) | 0.99 (0.92–1.00) | 0.93 (0.66–1.00) | 0.92 (0.83–0.97) |
| DiaSorin IgG | DiaSorin IgG | | | | | | | | | | |
| i Venous serum | Capillary serum | 82 | 22 (26.8%) | 21 (25.6%) | 0.97 (0.11) | <0.001 | 0.98 (0.93–1.00) | 0.96 (0.77–1.00) | 1.00 (0.94–1.00) | 1.00 (0.84–1.00) | 0.98 (0.91–1.00) |
| DiaSorin IgG | NADAL IgG | | | | | | | | | | |
| ii Venous Serum | Whole venous blood | 36 | 16 (44.4%) | 13 (36.1%) | 0.83 (0.16) | <0.001 | 0.91 (0.81–1.00) | 0.81 (0.54–0.96) | 1.00 (0.83–1.00) | 1.00 (0.75–1.00) | 1.00 (0.83–1.00) |
| iii Capillary Serum | Whole capillary blood | 58 | 15 (25.9%) | 11 (19.0%) | 0.80 (0.13) | <0.001 | 0.87 (0.75–0.98) | 0.73 (0.45–0.92) | 1.00 (0.92–1.00) | 1.00 (0.72–1.00) | 0.92 (0.80–0.98) |
| NADAL IgG +IgM | NADAL IgG +IgM | | | | | | | | | | |
| iv Venous serum | Whole capillary blood | 64 | 16 (25.0%) | 12 (18.8%) | 0.82 (0.12) | <0.001 | 0.77 (0.77–0.98) | 0.75 (0.48–0.93) | 1.00 (0.93–1.00) | 1.00 (0.74–1.00) | 0.92 (0.82–0.98) |
| v Venous serum | Whole venous blood | 36 | 15 (41.7%) | 13 (36.1 %) | 0.88 (0.17) | <0.001 | 0.93 (0.84–1.00) | 0.87 (0.69–0.98) | 1.00 (0.84–1.00) | 1.00 (0.75–1.00) | 0.91 (0.72–0.99) |
| vi Capillary serum | Whole capillary blood | 16 | 14 (87.5%) | 13 (81.3%) | 0.77 (0.24) | <0.001 | 0.96 (0.89–1.00) | 0.93 (0.66–1.00) | 1.00 (0.92–1.00) | 1.00 (0.75–1.00) | 0.67 (0.09–0.99) |

[a]Assay and blood source/substrate serving as the comparator.
[b]Assay and blood source/substrate compared to the reference.
[c]κ (SE), Cohen's kappa coefficient (standard error).
[d]P- values of kappa coefficient.
[e]Defined as the area under the receiving operating curve (AUROC).
[f]Positive predictive value.
[g]Negative predictive value.

this difference was not statistically significant ($P$ = 0.14), while the specificity remained perfect (Table 2). With five individuals testing positive only for IgM and four for both IgM and IgG on the NADAL test, the sensitivity analysis, including both IgG and IgM results from NADAL, maintained similar performance and concordance characteristics. The test performance shown in Table 2 is visually reflected in the violin plots (Fig. 2), which display the distribution, geometric mean, and range of DiaSorin IgG levels among NADAL positive and negative samples, demonstrating near-perfect group separation when NADAL was performed on venous serum samples, with clear visual dissociation between positive and negative results. For whole capillary blood, group separation appeared moderate, with no false positives and a few false negatives emerging in the 15–30 AU/mL range, a transitional zone where antibody levels were relatively low. These findings align with the logistic regression results (Fig. S3), which show an increase in predicted NADAL positivity with increase in DiaSorin IgG levels, particularly around the 15 AU/mL threshold.

As for secondary outcomes, the within-test agreement and performance of the DiaSorin on capillary serum was nearly identical to that on venous serum, with the latter being the manufacturer-recommended substrate. When comparing NADAL IgG on whole venous blood against DiaSorin IgG on venous serum, we observed a strong, yet lower agreement and high accuracy with slightly lower sensitivity (0.88 vs 0.81, $P$ = 0.30) than between NADAL IgG vs DiaSorin IgG, both measured on venous serum. When comparing NADAL IgG on whole capillary blood with DiaSorin IgG on capillary serum (same blood source), we observed a strong agreement (κ = 0.80) and moderate (0.87) accuracy, similar to that observed when comparing DiaSorin IgG on venous serum as one of the co-primary outcomes.

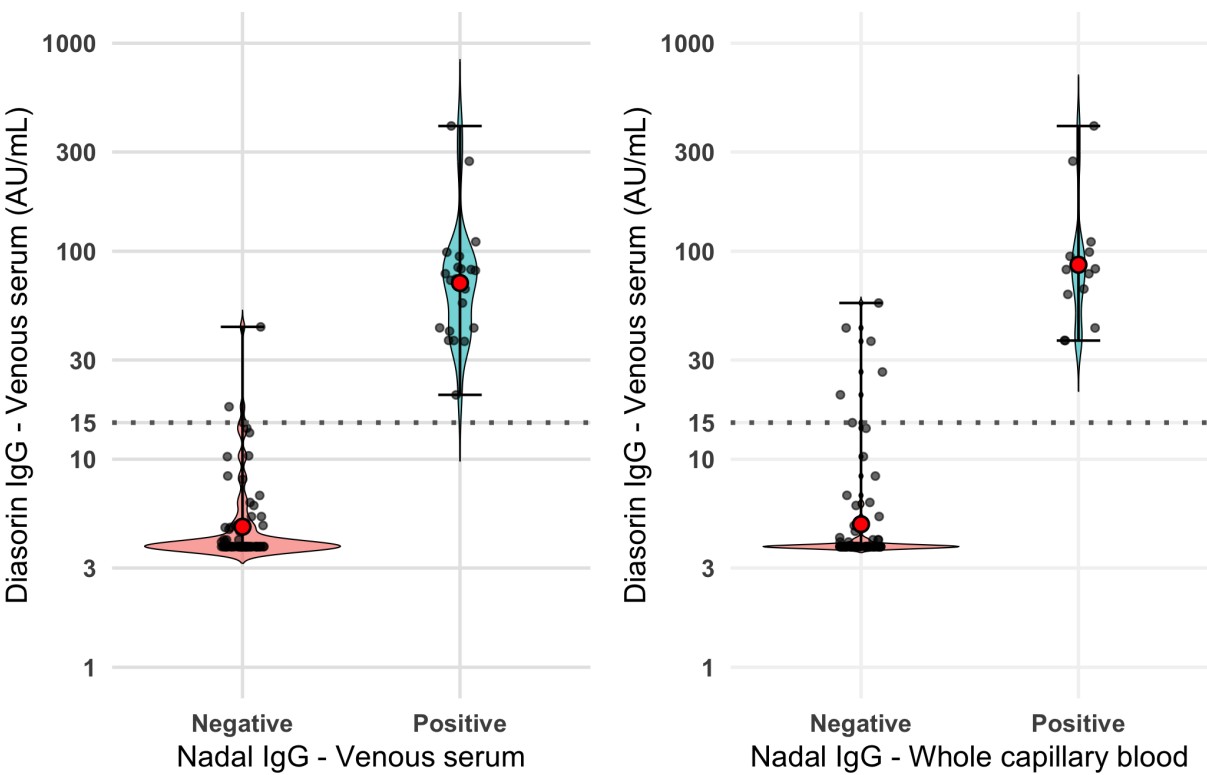

**FIG 2** Distribution of DiaSorin IgG levels in venous serum by Nadal IgG test result on venous serum and whole capillary blood. Both panels display two violin plots representing the distribution of the log-transformed DiaSorin IgG levels (AU/mL) in venous serum (y-axis), according to the NADAL IgG test results (positive or negative) on the x-axis, when analyzed in venous serum (left panel) and the whole capillary blood (right panel). Violin plots depict the geometric mean (red dot) distribution of IgG levels with individual data points overlaid and the range (black error bars). Each violin represents the density of DiaSorin values within each NADAL result group, where the width reflects the concentration of data points at specific antibody levels. Wider regions of the violins indicate a higher frequency of values at that level. Horizontal dotted lines indicate the 15 AU/mL DiaSorin threshold for seropositivity.

When assessing the within-test NADAL IgG and IgM agreement and performance across different blood substrates and sources, whole capillary blood vs venous serum exhibited strong agreement (κ = 0.82) but lower (0.77) accuracy and poorer (0.75) sensitivity. When comparing different substrates (whole blood vs serum) within the same blood source (capillary or venous), stronger (>90%) accuracy and sensitivity were observed. However, moderate, yet lower (κ = 0.77) agreement was observed in the small sample of 16 participants with 87.5% positivity prevalence comparing whole capillary blood to capillary serum, than observed between whole venous blood and venous serum), where a strong (κ = 0.88) agreement was observed, with a positivity rate of 41.7%.

## DISCUSSION

This study was conducted in the first year of the COVID pandemic and involved adults and children, in whom approximately a quarter had positive SARS-CoV-2 serology, as determined by the DiaSorin SARS-CoV-2 S1/S2 IgG on venous serum samples. Under these conditions, the NADAL COVID-19 IgG/IgM serology test assay, tested on the same blood source and substrate (venous serum), demonstrated near-perfect agreement, robust test performance, and group separation, including high accuracy, compared to the gold standard. This strong test performance was consistent whether evaluating IgG alone or both IgG and IgM assays from the NADAL test. However, the agreement and performance of the NADAL were lower when using both a different substrate and blood source than that of the DiaSorin, suggesting that blood source and/or substrate may be at play. Importantly, there was no impact on specificity as no false positives were observed.

Our co-primary outcomes focused on comparing the IgG serology from both the NADAL and DiaSorin tests to ensure a fair and consistent benchmark for assessing SARS-CoV-2 exposure. Most of the positive results on the NADAL test were due to IgG detection, which aligns with IgG as a stable marker of previous infections (29, 30). Notably, only a few participants showed positivity for IgM alone, which did not significantly impact the overall test performance comparisons. Whether focusing on IgG alone or combining IgG and IgM, the findings remained robust, highlighting the superior stability and reliability of IgG as a marker for long-term serological evaluation (31). Additionally, using venous serum as a substrate demonstrated high accuracy, sensitivity, and specificity of the NADAL vs DiaSorin tests, reinforcing its reliability as the preferred blood source and substrate for consistent test performance, as supported by the existing literature (32–34).

In this study, the NADAL test demonstrated near-perfect agreement with the DiaSorin assay using venous serum, highlighting strong performance under standardized approved conditions. This finding is consistent with those of several other studies evaluating assays on venous serum. For example, Nicol et al. reported high agreement (κ = 0.94) between CLIA detecting anti-N SARS-CoV-2 IgG and LFIA assays (33). Kahre et al. observed stable antibody levels across assays using venous serum, particularly with the DiaSorin platform (32). Similarly, Hackner et al. observed a strong agreement (97.7%) between the NADAL rapid test and ELISA in detecting IgG and IgM (34), while Domen et al. reported a high overall percentage agreement (95.9%) vs another LFIA (35). Garlantézec et al. reported a strong agreement (κ = 0.85) between a rapid LFIA and ELISA (36). These findings collectively underscore the reliability of venous serum as a blood source and substrate across different serological platforms for antigenic tests.

With regards to the second co-primary outcome, we observed a significantly lower (κ = 0.77 vs 0.91) between-assay agreement when whole capillary blood instead of venous serum was used as the NADAL substrate, compared to the gold standard tested on approved venous serum. These findings are consistent with those of existing literature. For example, Navarro et al. reported a lower sensitivity of 36.7% for whole capillary blood compared to 96.6% for venous serum (37), while both Mulder et al. and Moshe et al.

observed similar trends with lateral flow assays, with the sensitivity ranging from 69% to 77% depending on the assay (38, 39).

We thus sought to investigate whether different substrates and/or blood sources were at play and whether some assays were less robust than others to such sample differences. First, we compared the two assays within the same blood source. Agreement between DiaSorin and NADAL was slightly lower but still strong (κ ranging from 0.80 to 0.83) when NADAL was tested on whole blood, providing the same blood source (venous or capillary) as that used for DiaSorin. Second, to test the possibility of varying between-assay robustness to changes in substrate and/or blood source, we focused on their impact within the same assay. The DiaSorin test demonstrated consistent performance when comparing capillary serum to venous serum, highlighting its robustness across blood sources, using the same substrate (serum) for this assay; admittedly, different substrates were not tested on this assay. The NADAL performed similarly when comparing whole venous blood to venous serum, suggesting similar within-assay performance on different substrates (whole blood vs serum) using the same venous blood source. The latter findings are aligned with those of Hackner et al., who reported similar percent agreement when comparing NADAL on venous serum to ELISA on venous serum (97.7%) and NADAL on whole venous blood to ELISA on venous serum (96.2%) (34). However, while not statistically significant, the within-test agreement for NADAL appeared to decline when comparing whole capillary blood to capillary serum, suggesting perhaps a lower robustness of the test to differences in substrates (whole blood vs serum) when using capillary blood. Admittedly, the other performance metrics of this comparison remain excellent in this small sample of 16 participants, possibly due to a high anti-S prevalence exceeding 85%, such that firm conclusions cannot be drawn. Collectively, the whole blood substrate may more greatly affect NADAL's performance than blood sources; of note, the whole blood substrate was not tested for DiaSorin. One cannot rule out the possibility that certain assays, including LFIA, may be more sensitive than others to a difference in blood substrates, particularly when capillary, rather than venous, blood is the selected source. Until this is further investigated, it seems reasonable to favor use of venous blood for optimal assay performance.

Despite the variation in agreement, accuracy, and sensitivity, specificity remained high for all comparisons across assays, blood substrates, and sources, minimizing the risk of false positives. In other words, these serology assays are useful for ruling out acquired immunity when the test is negative, irrespective of blood substrates and sources. Their role in documenting immunity (with few false negatives) appears to be affected more by blood substrate than blood sources, particularly with capillary blood, thus limiting diagnostic utility depending on the aim of testing. This was reflected in the visual separation of groups, which was most distinct with venous serum, supporting its utility for a clear classification near the antibody threshold. In contrast, capillary blood showed more overlap around 15 AU/mL, consistent with reduced sensitivity in this range.

Our study has several limitations. Although we examined NADAL and DiaSorin's performance across blood and substrates, the small sample size in several comparisons, particularly those involving whole venous blood and whole capillary blood for NADAL, impacted the precision of our findings and led to wide confidence intervals. Given early-pandemic constraints, our sample size was limited by severe logistical barriers (e.g., restriction of travel to the research site for venipuncture and the time-sensitive need to initiate our COVID trials). Yet our findings are now applicable to the new appeal of remote research visits and blood sampling. Despite our evaluation of NADAL's test performance across multiple substrates and blood sources, we did not assess DiaSorin's performance across different substrates (only across blood sources using serum), preventing firm conclusions about DiaSorin's robustness for alternate substrates. Similarly, we did not directly compare NADAL's performance across blood sources for the same substrate (serum), limiting conclusions about its consistency under such conditions. Our study focused solely on IgG detection using qualitative assays that did not assess neutralization capacity. Building on our initial need to screen for past infection

in immunocompetent individuals, our study was designed to evaluate test concordance and identify conditions that affect test accuracy.

The findings should be interpreted in the context of the study period, with a relatively high prevalence of acquired immunity due to infection before mass vaccination and when the main application was to identify previously infected individuals, those lacking vaccination documentation. Caution should be exercised when extrapolating our findings to assays detecting anti-N antibodies to ascertain the immune status in highly vaccinated populations as our assays only detected anti-S antibodies. Since the end of the pandemic, the identification of the SARS-CoV-2 immune status of an individual or population is much less relevant. However, with new outbreaks, serology tests will again play an important role in specific settings, particularly to assess the immune status in individuals without access to antigen testing, vaccinated immunosuppressed individuals, and epidemiological surveillance.

## Conclusion

The NADAL COVID-19 IgG/IgM serology test demonstrated excellent performance, with near-perfect agreement and high accuracy compared to the DiaSorin Liaison SARS-CoV-2 S1/S2 IgG assay when tested on the same approved blood sources and substrates (venous serum). Within-test agreement and performance were relatively well-preserved when applied to different blood sources (capillary or venous), provided the substrate (serum) remained the same. However, the performance appeared to decline when the test was applied to different substrates, particularly whole capillary blood, as used in point-of-care settings without prior processing. While NADAL maintained high specificity across all conditions, its reduced sensitivity on whole capillary blood underscores limitations for certain diagnostic applications. These findings highlight the necessity of rigorous testing and validation of new serology assays to ensure consistent performance before use on different substrates and sources than those approved, ensuring trustworthy diagnostic results.

### ACKNOWLEDGMENTS

We acknowledge the infrastructure support of the Fonds de la Recherche du Québec en Santé (FRQS) provided to the Azrieli Research Institute of CHUSJ. We thank participants enrolled in the VALICAP and PROTECT studies. We thank Jing Leng for her assistance with coordinating and enrolling the participants.

This work was funded by a grant awarded through a peer-reviewed process of the COVID-19 May 2020 Rapid Response Funding Opportunity by the Canadian Institute of Health Research (grant number 447317)

F.M.D. designed, obtained funding, and supervised the conduct, analysis, and interpretation of data. H.D. participated in recruitment and data collection. H.D. performed the statistical analysis with input from F.M.D. H.D. wrote the first draft of the article with input from all authors. All authors have read and approved the published version of the manuscript.

### AUTHOR AFFILIATIONS

[1]Clinical Research and Knowledge Transfer Unit on Childhood Asthma, Centre Hospitalier Universitaire Sainte-Justine, Montreal, Québec, Canada

[2]Departments of Biomedical Sciences, Faculty of Medicine, Universite de Montreal, Montreal, Québec, Canada

[3]Optilab Montréal-Sainte-Justine, Departments of Pediatrics and Microbiology, Infectiology and Immunology Centre, Hospitalier Universitaire Sainte-Justine, Montreal, Québec, Canada

[4]Microbiology and Infectious Disease, Centre Universitaire de santé de Montréal (CHUM), University of Montreal, Montreal, Québec, Canada

[5]Departments of Pediatrics and of Social and Preventive Medicine, University of Montréal, Montreal, Québec, Canada

## AUTHOR ORCIDs

Harika Dasari ⬤ http://orcid.org/0000-0002-0979-0956

## FUNDING

| Funder | Grant(s) | Author(s) |
|---|---|---|
| Canadian Institutes of Health Research | 447317 | Francine M. Ducharme |

## AUTHOR CONTRIBUTIONS

Harika Dasari, Conceptualization, Data curation, Formal analysis, Methodology, Project administration, Visualization, Writing – original draft | Claude Bourassa, Supervision, Writing – review and editing | Christian Renaud, Writing – review and editing | Cécile Tremblay, Writing – review and editing | Francine M. Ducharme, Conceptualization, Funding acquisition, Investigation, Methodology, Project administration, Supervision, Visualization, Writing – review and editing

## DATA AVAILABILITY

A de-identified dataset supporting the findings of this study is publicly available in the Harvard Dataverse: https://doi.org/10.7910/DVN/HZ2EBX.

## ADDITIONAL FILES

The following material is available online.

### Supplemental Material

**Supplemental Figures (Spectrum03350-24-S0001.docx).** Fig. S1 to S3.

### Open Peer Review

**PEER REVIEW HISTORY (review-history.pdf).** An accounting of the reviewer comments and feedback.

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
