## [Reviewer comments · Microbiology Spectrum]

Microbiology Spectrum

Comparison of NADAL COVID IgG/IgM rapid test and DiaSorin Liaison SARS-CoV-2 S1/S2 IgG assay across different blood sources and substrates.

Harika Dasari, Claude Bourassa, Christian Renaud, Cecile Tremblay, and Francine Ducharme

Corresponding Author(s): Harika Dasari, Universite de Montreal

Review Timeline:

Submission Date:	January 11, 2025
Editorial Decision:	March 2, 2025
Revision Received:	April 6, 2025
Accepted:	April 22, 2025

Editor: Chuan Lim

Reviewer(s): Disclosure of reviewer identity is with reference to reviewer comments included in decision letter(s). The following individuals involved in review of your submission have agreed to reveal their identity: sara assadiasl (Reviewer #2)

Transaction Report:

DOI: <https://doi.org/10.1128/spectrum.03350-24>

Re: Spectrum03350-24 (Comparison of NADAL COVID IgG/IgM rapid test and DiaSorin Liaison SARS-CoV-2 S1/S2 IgG assay across different blood sources and substrates.)

Dear Dr. Harika Dasari:

Thank you for the privilege of reviewing your work. Below you will find my comments, instructions from the Spectrum editorial office, and the reviewer comments.

Revision Guidelines

Sincerely,
Chuan Lim
Editor
Microbiology Spectrum

Reviewer #1 (Comments for the Author):

The manuscript "Comparison of NADAL COVID IgG/IgM rapid test and DiaSorin Liaison SARS-CoV-2 S1/S2 IgG assay across different blood sources and substrates." compares whole blood to plasma as well as a new lateral flow test to a chemiluminescent immunoassays. The strategies are straightforward but lack analysis at low antibody levels. The correlation will be high at samples with high antibody levels. If a high percent of samples have high antibody levels the correlation will be

high. However if a high number of samples have low antibody levels the correlation will be lower. The discussion needs to be amended that the correlations are antibody level dependent and will vary based on population level of SARS-CoV-2 antibodies. What is correlation of upper third of antibody levels, middle third, and lower third amongst the two tests and blood versus plasma? If diasorin uses up to 400 AU what is correlation 0-100, 101 to 250 and 250 to 400 in diasorin units to the diverse tests.

There are places in the manuscript that blur antigen or nucleic acid detection for acute viral infection with antibody based diagnosis which is more a concern in the 3% of the population which are immunosuppressed. Not sure of the diagnostic utility translating to management decision in immunocompetent people.

For instance in line 65 and 66 what is the point of a yes not antibody test for the general population. What is the purpose of self-screening serology unless you want to see if you responded to vaccination if immunosuppressed. High low antibody levels do not have much meaning unless tied to neutralizations.

Line 73 why do we need quick convenient lateral flow antibody tests when there has been a significant drop off in any antibody test except for qualification of blood donors for COVID-19 convalescent plasma.

This is a look back to relevancy 4 years ago when lateral flow might have been useful on samples from 4 years ago.

The work looks at about 80 to 90 samples for both test comparison.

Line 189 is an antibody level comparison as samples have high levels of antibodies and will correlate. Low levels of antibodies do not correlate as well.

Discussion

Repeats a lot of previous stated results.

Needs an antibody level correlation discussion and why does this matter now?

Needs a graph on range, geometric mean and all points for all samples in the diasorin test

Reviewer #2 (Comments for the Author):

The article "Comparison of NADAL COVID IgG/IgM rapid test and DiaSorin Liaison SARS-CoV-2 S1/S2 IgG assay across different blood sources and substrates" reports the validity of a new rapid COVID test (NADAL) compared to a standard test (DiaSorin).

Regarding the content:

It is very helpful to design accurate and rapid POC tests for COVID, and there have been many suggestions. Although the findings are interesting, I am not sure about the utility of the results because the study was conducted between 2020 and 2021 when mass vaccination had not been performed globally. Today, everybody might have anti-COVID IgG in blood circulation due to the vaccination or infection, and as far as I saw in Figure S1, the positive IgM line of NADAL kit was very faint and unreliable; therefore, its usefulness for detecting new infections (indicated by IgM+ test) is doubted.

Regarding the form:

I appreciate the efforts of the authors to conduct thorough research; however, the study design and reporting style are confusing and hard to follow. It might be better to provide separate result sections, each one describing a distinct part of the comparison (sample sources, sample substrates, IgG, IgM, etc.). In addition, the sample size is too low for today. The age report in Table 1 needs to be corrected (mean {plus minus} SD).

The article “Comparison of NADAL COVID IgG/IgM rapid test and DiaSorin Liaison SARS-CoV-2 S1/S2 IgG assay across different blood sources and substrates” reports the validity of a new rapid COVID test (NADAL) compared to a standard test (DiaSorin).

Regarding the content:

It is very helpful to design accurate and rapid POC tests for COVID, and there have been many suggestions. Although the findings are interesting, I am not sure about the utility of the results because the study was conducted between 2020 and 2021 when mass vaccination had not been performed globally. Today, everybody might have anti-COVID IgG in blood circulation due to the vaccination or infection, and as far as I saw in Figure S1, the positive IgM line of NADAL kit was very fade and unreliable; therefore, its usefulness for detecting new infections (indicated by IgM+ test) is doubted.

Regarding the form:

I appreciate the efforts of the authors to conduct thorough research; however, the study design and reporting style are confusing and hard to follow. It might be better to provide separate result sections, each one describing a distinct part of the comparison (sample sources, sample substrates, IgG, IgM, etc.). In addition, the sample size is too low for today. The age report in Table 1 needs to be corrected (mean \pm SD).

Reviewer #1

Comment #1: *“The strategies are straightforward but lack analysis at low antibody levels. The correlation will be high at samples with high antibody levels. If a high percent of samples have high antibody levels the correlation will be high. However if a high number of samples have low antibody levels the correlation will be lower. The discussion needs to be amended that the correlations are antibody level dependent and will vary based on population level of SARS-CoV-2 antibodies. What is correlation of upper third of antibody levels , middle third, and lower third amongst the two tests and blood versus plasma? If diasorin uses up to 400 AU what is correlation 0-100, 101to 250 and 250 to 400 in diasorn units to the diverse tests?”*

We thank the reviewer for this thoughtful comment. We agree that test performance and correlation are influenced by the underlying distribution of antibody levels. In fact, the correlation between two continuous variables is driven not only by the proximity of the data points to the regression line but also by the extreme values. In this regard, we observe values varying from 15 AU/mL to 50 AU/mL. However, the NADAL test yields a binary result, and its performance is typically interpreted in reference to the DiaSorin threshold of 15 AU/mL which defines seropositivity. While we considered computing correlation coefficients within DiaSorin IgG tertiles or defined ranges (e.g., 0–100, 101–250, 251–400 AU/mL), this approach would not provide clinically meaningful insight as stratification into arbitrary DiaSorin ranges above or below this threshold would not reflect the test’s clinical application. Instead, we used logistic regression (Figure S3) to model the probability of NADAL IgG positivity across the continuous DiaSorin range, which better captures the transition zone near the threshold. We also included violin plots (Figure 2) to show the distribution, range, and geometric mean of DiaSorin values by NADAL result. These approaches provide clearer interpretability while avoiding arbitrary stratification. These results confirm the threshold-dependent nature of NADAL test performance and are now described in both the Results (2nd paragraph) and Discussion (1st and 6th paragraph) sections.

Comment #2: *“There are places in the manuscript that blur antigen or nucleic acid detection for acute viral infection with antibody-based diagnosis, which is more of a concern in the 3% of the population that is immunosuppressed. Not sure of the diagnostic utility translating to management decisions in immunocompetent people”*

We appreciate this important distinction. We acknowledge that positive antibody-based serological tests imply immunocompetence and are not intended for diagnosing acute SARS-CoV-2 infection. Our study focused on evaluating the likelihood of prior exposure through antibody detection in an immunocompetent population. Immunosuppressed individuals were excluded from our population (we clarified this in the selection criteria). We have revised the Introduction (1st paragraph) and Discussion (7th paragraph) to clearly state that the NADAL and DiaSorin IgG assays were used to assess past immune response, not active infection, in this population. In the discussion, we clarified that our findings apply to immunocompetent individuals and highlight the continued relevance of antibody testing for epidemiological surveillance and post-infection monitoring in this population. We also reviewed the manuscript to ensure clear differentiation between antibody- and antigen-based diagnostics, making adjustments where needed to prevent confusion.

Comment #3: *"For instance, in line 65 and 66, what is the point of a yes/no antibody test for the general population? What is the purpose of self-screening serology unless you want to see if you responded to vaccination if immunosuppressed? High or low antibody levels do not have much meaning unless tied to neutralization capacity."*

We thank the reviewer for this insightful comment. We agree that antibody testing does not inform acute infection and that antibody levels are only partially informative without neutralization data. In our study, the purpose of using point-of-care antibody tests was to assess seroprevalence in an immunocompetent population, not to guide individual-level clinical decisions. We have clarified that neutralization capacity was not measured and is outside the scope of this study. The revised Introduction (1st paragraph) and Discussion (7th paragraph) now distinguish serology from antigen testing and emphasize that while routine self-screening is no longer recommended, qualitative antibody tests remain valuable for population-level surveillance and assessing prior immune response in specific settings.

Comment#4: *"Line 73: Why do we need quick, convenient lateral flow antibody tests when there has been a significant drop-off in any antibody testing except for qualification of blood donors for COVID-19 convalescent plasma?"*

We appreciate this perspective and adjusted our framing of the relevance of the lateral flow antibody test. Discussion (7th-8th paragraphs) now clarifies that the demand for antibody testing has decreased over time as vaccination campaigns have progressed. However, we emphasize that lateral flow assays still have utility in specialized settings, particularly: resource-limited settings where laboratory-based serological testing is unavailable, and field epidemiology studies where rapid screening can help assess past exposure patterns and immunosuppressed patients, where assessing immune response post-vaccination remains relevant. At the time of the study (Fall 2020), access to qPCR testing had been extremely limited from the beginning of the pandemic and rapid antigen tests were not yet on the market; the only scientifically valid approach to confirm prior infection was by serology testing.

Comment#5: *"This is a look back to relevancy four years ago when lateral flow might have been useful on samples from four years ago."*

Yes, We acknowledge that widespread public use of serology tests has declined since the early pandemic phase. However, we argue that the scientific relevance of evaluating serological assays remains intact because: Understanding assay performance across different blood substrates remains a methodological priority, independent of immediate public health demands. Serology assays continue to play a role in monitoring immune response in vulnerable populations, including immunosuppressed individuals and those undergoing post-vaccination assessments. Future applications of similar lateral flow technologies extend beyond COVID-19, making performance evaluations valuable for broader infectious disease research. To reflect this, we revised the Discussion (7th-8th paragraphs) to contextualize the study's findings in the current landscape while maintaining their scientific merit.

Comment#8: *"Line 189: The antibody level comparison is biased because samples have high levels of antibodies and will correlate. Low levels of antibodies do not correlate as well."*

We appreciate the reviewer's observation and agree that correlation should be better when there are high antibody levels. We now provide analyses incorporating the full range of DiaSorin IgG values to better ascertain the concordance over the whole spectrum of antibody level with the violin plots (Figure 2) and logistic regression model (Figure S3), illustrating test behaviour across this spectrum, including the transition around the 15 AU/mL threshold.

Reviewer 2:

Comment #1: *“Although the findings are interesting, I am not sure about the utility of the results because the study was conducted between 2020 and 2021 when mass vaccination had not been performed globally. Today, everybody might have anti-COVID IgG in blood circulation due to vaccination or infection.”*

We acknowledge this concern and have revised the Discussion section (1st paragraph) to explicitly state that this study was conducted early in the pandemic (2020-early 2021) before widespread vaccination. We have emphasized that the findings are relevant in specific settings, such as assessing past infection status in immunocompetent individuals or for epidemiological surveillance of population immune status. These clarifications ensure that the study's relevance is framed appropriately given current public health contexts.

Comment#2: *“As far as I saw in Figure S1, the positive IgM line of the NADAL kit was very faint and unreliable; therefore, its usefulness for detecting new infections (indicated by IgM+ test) is doubted.”*

We thank the reviewer for this observation. The faint line is not necessarily representative of all positive IgM tests. Some results show a very clear line. Importantly, the manuscript clarifies that the inclusion of IgM did not meaningfully alter overall test performance compared to IgG alone; the overwhelming majority had past rather than very recent infection (Table 2). As noted in the Discussion (2nd Paragraph), the findings remained robust whether evaluating IgG alone or in combination with IgM, reinforcing the superior stability of IgG as a marker for serological evaluation. Given that IgM detection was not a primary focus of this study and contributed minimally to overall positivity, we have not expanded on variability related to occasional faint IgM bands in the manuscript.

Comment #3: *“I appreciate the efforts of the authors to conduct thorough research; however, the study design and reporting style are confusing and hard to follow. It might be better to provide separate result sections, each one describing a distinct part of the comparison (sample sources, sample substrates, IgG, IgM, etc.).”*

We appreciate the reviewer's suggestions to increase the clarity of reporting. In response, we have revised the *Results* section to provide a more structured and accessible presentation of the study findings. The section is now organized into clear paragraphs, each dedicated to a specific aspect of the comparison: (1) the primary comparison of NADAL versus DiaSorin using venous serum; (2) comparisons across different blood sources (venous vs. capillary); (3) comparisons across different substrates (serum vs. whole blood); and (4) evaluation of IgG versus IgM

performance. This reorganization enhances readability and ensure that each analytical component is distinctly presented for the reader.

Comment #4: *“In addition, the sample size is too low for today.”*

We thank the reviewer for this important comment and the opportunity to clarify. Given the logistical constraints of the early COVID-19 pandemic (2020–2021), particularly the challenge of obtaining paired venous and capillary samples, we have clarified in the revised manuscript that this study was designed as a proof-of-concept evaluation of serological test performance across blood sources and substrates. At the time, the NADAL test had only been approved for use on venous serum, and this study served to evaluate its broader utility in immunocompetent individuals. The relatively small sample size was also shaped by the urgency of launching a larger clinical trial (PROTECT), in which this test was used to exclude individuals with prior infection from contributing to the primary composite endpoint. As such, the early results informed the practical implementation of NADAL in a real-world screening context. We also respectfully note that the sample size was sufficient to achieve statistical significance for all agreement estimates (all P values < 0.001) and allowed for relatively tight confidence intervals ($\pm 10\%$) for most accuracy metrics. However, we recognize that for estimates falling below 0.90, particularly sensitivity in the capillary blood subgroup, the limited power led to wider confidence intervals. We have added this nuance in the discussion to contextualize the findings and reinforce the value of cautious interpretation.

Comment #5: *“The age report in Table 1 needs to be corrected (mean {plus minus} SD).”*

We appreciate this observation. This was an oversight on our part as age was not normally distributed in our sample. Therefore, we have updated Table 1 to report age as the median with interquartile range (IQR: 25th–75th percentile), which is the appropriate measure for skewed data. This correction ensures that the data are accurately represented in line with epidemiological best practices.

Closing Statement: We sincerely appreciate the thoughtful feedback from both reviewers and the Editor. We believe the requested revisions will substantially strengthen the manuscript and ensure it is clear, methodologically transparent, and broadly relevant beyond the immediate context of COVID-19.

Re: Spectrum03350-24R1 (Comparison of NADAL COVID IgG/IgM rapid test and DiaSorin Liaison SARS-CoV-2 S1/S2 IgG assay across different blood sources and substrates.)

Dear Dr. Harika Dasari:

Your manuscript has been accepted, and I am forwarding it to the ASM production staff for publication. Your paper will first be checked to make sure all elements meet the technical requirements. ASM staff will contact you if anything needs to be revised before copyediting and production can begin. Otherwise, you will be notified when your proofs are ready to be viewed.

Sincerely,
Chuan Lim
Editor
Microbiology Spectrum

Reviewer #1 (Comments for the Author):

Thanks for adding fig 2 and sfig 3
comments addressed to the best possible manner with limited number of samples

Reviewer #2 (Comments for the Author):

no comment